# A General Unified Graph Neural Network Framework Against Adversarial Attacks

## Abstract

Graph Neural Networks (GNNs) are powerful tools in representation learning for graphs. However, they are reported to be vulnerable to adversarial attacks, raising numerous concerns for applying it in some risk-sensitive domains. Therefore, it is essential to develop a robust GNN model to defend against adversarial attacks. Existing studies address this issue only considering cleaning perturbed graph structure, and almost none of them simultaneously consider denoising features. As the graph and features are interrelated and influence each other, we propose a General Unified Graph Neural Network (GUGNN) framework to jointly clean the graph and denoise features of data. On this basis, we further extend it by introducing two operations and develop a robust GNN model(R-GUGNN) to defend against adversarial attacks. One operation is reconstructing the graph with its intrinsic properties, including similarity of two adjacent nodes' features, sparsity of real-world graphs and many slight noises having small eigenvalues in perturbed graphs. The other is the convolution operation for features to find the optimal solution adopting the Laplacian smoothness and the prior knowledge that nodes with many neighbors are difficult to attack. Experiments on four real-world datasets demonstrate that R-GUGNN has greatly improved the overall robustness over the state-of-the-art baselines.

## 1 Introduction

Graph Neural Networks(GNNs) have drawn great attention as graphs can represent complex relationships among nodes. Graphs are ubiquitous in different domains, which are usually applied in recommender systems(Ying et al., 2018a), chemistry(Duvenaud et al., 2015), social media(Qiu et al., 2018) and so on. Utilizing the strong representation capacity of graphs, we can enhance performance of down-stream tasks such as node classification(Kipf & Welling, 2017; Velickovic et al., 2018; Klicpera et al., 2019), link prediction(Grover & Leskovec, 2016; Bojchevski et al., 2018) and graph classification(Defferrard et al., 2016; Ying et al., 2018b). A GNN model often consists of several graph convolution layers. A common practice of convolution layers is utilizing a feed-forward network to transform features and then aggregating transformed features. A series of convolution layers have been proposed and achieved great success such as GCN(Kipf & Welling, 2017), GAT(Velickovic et al., 2018) and PPNP(Klicpera et al., 2019).

However, GNN models composed of these convolution layers are vulnerable to adversarial attacks. Attacks can be conducted on either node features or the graph structure, while most existing adversarial attacks on graph data focus on modifying the graph structure(Xu et al., 2020). They always try to add, delete, or rewire edges to change the graph structure. Although these perturbations are unnoticeable, they can easily degrade the performance of GNN models, which may cause bad consequences. For example, spammers may create virtual followers to increase the chance of false messages being recommended and spread. The lack of GNNs' robustness raises increasing concerns for applying it in some risk-sensitive domains. Therefore, it is necessary to develop graph defense techniques. Many existing defense methods focus on cleaning perturbed graphs by detecting properties of clean graphs and effects of specific attacks on graphs(Entezari et al., 2020; Jin et al., 2020b). Prior knowledge according to these researches can help GNN models defend against adversarial attacks to a certain extent. The study(Jin et al., 2020b) has proved that adversarial attacks could lead perturbed graphs to violate some properties of real graphs. For example, the rank of attacked graph increases and adversarial attacks often connect nodes with large feature differences.

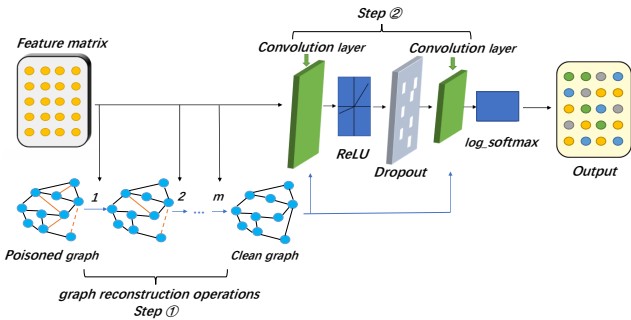

Figure 1: Concrete design of R-GUGNN. We clean the attacked graph and denoise features.

However, existing studies only focus on cleaning the perturbed graph structure, and almost none of them simultaneously consider denoising features.

As the graph and features are closely tied and contain perturbations and noises, in this paper, we propose a General Unified Graph Neural Network(GUGNN) framework to jointly clean the graph and denoise features. Based on the GUGNN framework, we further introduce two kinds of operations. One operation is reconstructing the graph with its properties that real-world graphs are sparse(Zhou et al., 2013), the features of two adjacent nodes tend to be similar(McPherson et al., 2001) and perturbed graphs have many slight noises with small eigenvalues. Nodes with more neighbors are hard to attack(Zügner et al., 2018). Though we cannot change the number of nodes' first-order neighbors, we can adjust the size of nodes' neighborhood to add some high-order neighbors to nodes. According to this principle, from the denoising perspective, we design the convolution operation for features to find the optimal solution. Utilizing the two kinds of operations above, we develop a robust model(R-GUGNN), which can be applied for defending against different adversarial attacks.

The contributions of this paper are summarized as follows:

- We propose GUGNN framework to jointly clean the graph and denoise features for perturbations and noises existing in the graph and features.
- We introduce two kinds of operations to clean attacked graphs and denoise features respectively based on the GUGNN framework.
- For defending against adversarial attacks, we develop a concrete model R-GUGNN to realize the goal of the GUGNN framework utilizing the two kinds of operations.
- Experiments show that R-GUGNN has a strong capacity for defending against different adversarial attacks and stably outperforms the state-of-the-art defense models.

## 2 RELATED WORK

This section has two parts, including graph neural networks, and adversarial attacks and defenses for GNNs.

### 2.1 GRAPH NEURAL NETWORKS

In this subsection, we review some famous graph neural network models, including GCN, GAT, PPNP, and a unified GNN framework UGNN. For more knowledge about GNNs, you can refer to some reviews(Wu et al., 2020; Zhang et al., 2020).

The convolution of GCN(Kipf & Welling, 2017) is defined in the graph spectral domain. Avoiding computing the full eigenvectors of the graph Laplacian matrix, based on Chebyshev polynomials, GCN only uses the first-order polynomial to simplify the graph convolution, which has been an accepted good graph convolution layer for its performance. The convolution of GAT(Velickovic et al., 2018) is defined in the spatial domain. The difference between GAT and GCN is that GAT learns different attention scores for neighbors when aggregating features. PPNP(Klicpera et al.,

2019) derives a propagation scheme based on personalized PageRank. It propagates information from a large and adjustable neighborhood instead of the first-order neighborhood directly. The neighborhood can be adjusted via a hyper-parameter.

UGNN(Ma et al., 2020) is a unified GNN framework available for different feature propagation processes from a denoising perspective. It proposes a denoising optimization problem with the Laplacian regularization term and tries to solve it in different ways utilizing the first derivative or the optimal solution. Original Laplacian matrix can be also replaced with different normalized forms. Different solutions to the optimization problem are corresponding to various convolution layers with different feature aggregation processes such as GCN, GAT, PPNP and so on.

## 2.2 ADVERSARIAL ATTACKS AND DEFENSES FOR GNNs

We recommend a repository DeepRobust(Li et al., 2020) for readers. It contains many adversarial attacks and defenses on the graph, which is quite useful for researchers. For more knowledge about adversarial attacks and defenses for GNNs, you can refer to the review(Jin et al., 2020a).

Some adversarial attack methods have been proposed to show the vulnerability of GNNs with some unnoticeable perturbations added to the graph structure or node attributes. In the field of node classification, the aim of adversarial attacks is fooling GNNs into classifying nodes incorrectly. Poisoning attacks change the graph structure before we train GNN models, which is one of the most common settings of adversarial attacks on graph data. Poisoning attacks have various types, including global attack, targeted attack and random attack. Based on the whole graph, the goal of global attack is to degrade the overall performance of GNNs. One of the state-of-the-art global attacks is $metattack$(Zügner & Günnemann, 2019), which generates the poisoning attacks based on meta-learning. Targeted attack generates attacks on some specific nodes and aims to fool GNNs on these target nodes. The $nettack$(Zügner et al., 2018) is one of the state-of-the-art targeted attacks, which aims to change the graph structure and features of target nodes or nearby nodes with perturbations remaining unnoticeable. Random attack adds random noises to the clean graph whose concrete practice is adding, removing or flipping edges randomly.

Methods about preventing GNNs from adversarial attacks are also developed to improve the robustness of GNNs recently. To mitigate the effects of adversarial attacks on the graph, RGCN(Zhu et al., 2019) uses Gaussian distributions as hidden representations of nodes instead of plain vectors in other GNNs. Considering that $nettack$ is a high-rank attack, GCN-SVD(Entezari et al., 2020) is proposed to reconstruct the perturbed graph with only the top-$k$ largest singular components. Using such a low-rank approximation, GCN-SVD can reduce the effects of $nettack$. In fact, the practice of ensuring low-rank is removing noises with small singular values of the graph. Pro-GNN(Jin et al., 2020b) jointly optimizes a structural graph and a robust GNN model from the perturbed graph with some properties of clean graphs. Pro-GNN has big improvement over other defense models with these properties. However, like other defense models, Pro-GNN doesn't take denoising features into account jointly.

## 3 THE PROPOSED FRAMEWORK

In this section, we first present GUGNN framework, and then we introduce a novel graph reconstruction operation. At last, we show the convolution operation for features and our concrete design of R-GUGNN model, which is used to realize the goal of GUGNN framework.

### 3.1 NOTIONS

We denote some notations here. Denote $\mathbf{X} \in R^{N \times d}$ as the feature matrix, where $N, d$ represent the number of samples and the dimension of features respectively. Denote $\mathcal{G} = \{\mathcal{V}, \mathcal{E}\}$ as the graph, where $\mathcal{V}$ represents node sets and $\mathcal{E}$ represents edge sets. We also use the adjacency matrix $\mathbf{A} \in R^{N \times N}$ to represent $\mathcal{G}$.

---

**Algorithm 1:** R-GUGNN

---

**Input:** Adjacency matrix $\mathbf{A}$, Feature matrix $\mathbf{X}$, Labels $y$, Hyper-parameters $m, c, \beta, \lambda$, Learning
      rate $\eta$.
**Output:** GNN parameters $\theta$

1   $\widetilde{\mathbf{A}} = \mathbf{A} + \mathbf{I}$; Initialize $\mathbf{S} = \widetilde{\mathbf{A}}$;
2   **for** $i$= 1 to $m$ **do**
3      |   Calculating $\mathbf{Z}$ in formula(5);
4      |   $\mathbf{S} = \mathbf{S} - \frac{c}{2}\mathbf{Z}$;
5      |   $\mathbf{S} = \text{prox}_{*\beta}(\mathbf{S})$;
6      |   $\mathbf{S} = \text{prox}_s(\mathbf{S})$;
7   Initialize $\mathbf{F} = \mathbf{X}$;Randomly initialize $\theta$;
8   **while** *Stopping condition is not met* **do**
9      |   Forward propagation using two convolution layers:
10     |   Using feature transformation formula: $\mathbf{F} = \mathbf{FW}$,
11     |   and feature aggregation formula: $\mathbf{F} = (\mathbf{I} + \lambda\hat{\mathbf{L}})^{-1}\mathbf{F}$;
12     |   Getting output $y'$;
13     |   Calculating gradient $g$ according to $y$ and $y'$;
14     |   Backward propagation: $\theta = \theta - \eta g$;
15   return $\theta$;

---

### 3.2 THE GENERAL UNIFIED GNN FRAMEWORK

To discard perturbations and noises in the graph and features, considering the tight connection between them, we propose our general united graph neural network(GUGNN) framework to solve such problem, which is shown as follows:

$$\underset{\mathbf{S},\mathbf{F}}{argmin}\,\mathcal{L} = \|\mathbf{S} - \mathbf{A}\|_F^2 + \gamma\|\mathbf{F} - \mathbf{X}\|_F^2 + c \cdot tr(\mathbf{F}^T\mathbf{LF}) + \beta \cdot f(\mathbf{S}) \tag{1}$$

where $\mathbf{S}$ and $\mathbf{F}$ are the learned adjacency and feature matrix. $\mathbf{L}$ is the Laplacian matrix of $\mathbf{S}$. $\mathbf{L} = \mathbf{D} - \mathbf{S}$, where $\mathbf{D}$ is a diagonal matrix and $\mathbf{D}_{ii} = \sum_{j=1}^N \mathbf{S}_{ij}$. $\mathbf{L}$ can be also replaced with different normalized forms. $tr(\mathbf{F}^T\mathbf{LF})$ is Laplacian regularization term for both denoising features and cleaning the graph. $f(\mathbf{S})$ is a flexible regularization term to enforce some prior over $\mathbf{S}$. $\gamma$, $c$ and $\beta$ are hyper-parameters to balance different components.

From a united perspective, we view that both features and the graph contain noises and our goal is jointly optimizing $\mathbf{F}$ and $\mathbf{S}$. $tr(\mathbf{F}^T\mathbf{LF})$ can be rewritten as $\frac{1}{2}\sum_{i,j=1}^N \mathbf{S}_{ij}(\mathbf{f}_i - \mathbf{f}_j)^2$, where $\mathbf{f}_i$ is the $i$-$th$ row of $\mathbf{F}$. This term represents that features of two adjacent nodes should be similar, which is the guidance for both learning $\mathbf{F}$ and $\mathbf{S}$. Although $\mathbf{X}$ and $\mathbf{A}$ have some noises, they can represent the real features and the graph to a large extent. So, the learned $\mathbf{F}$ and $\mathbf{S}$ should be similar to $\mathbf{X}$ and $\mathbf{A}$ respectively, which are the meanings of $\|\mathbf{F} - \mathbf{X}\|_F^2$ and $\|\mathbf{S} - \mathbf{A}\|_F^2$. In addition, we add some prior to the graph in $f(\mathbf{S})$ to make it more accurate.

### 3.3 THE NOVEL GRAPH RECONSTRUCTION OPERATION

We focus on cleaning the perturbed graph supposing $\mathbf{F}=\mathbf{X}$. Formula(1) of GUGNN can be rewritten as follows:

$$\underset{\mathbf{S}}{argmin}\,\mathcal{L} = \|\mathbf{S} - \mathbf{A}\|_F^2 + c \cdot tr(\mathbf{X}^T\mathbf{LX}) + \beta \cdot f(\mathbf{S}) \tag{2}$$

Considering that the graph contains noises, we rewrite formula(2) as follows:

$$\underset{\mathbf{S}}{argmin}\,\mathcal{L} = \|\mathbf{S} - \widetilde{\mathbf{A}}\|_F^2 + c \cdot tr(\mathbf{X}^T\hat{\mathbf{L}}\mathbf{X}) + \beta\|\mathbf{S}\|_*$$
$$= \mathcal{L}1 + \mathcal{L}2 + \mathcal{L}3 \tag{3}$$

The adjacency matrix with self-loop $\widetilde{\mathbf{A}}$ and the normalized Laplacian matrix $\hat{\mathbf{L}}$ are adopted. $\hat{\mathbf{L}} = \mathbf{D}^{-\frac{1}{2}}\mathbf{L}\mathbf{D}^{-\frac{1}{2}}$. $tr(\mathbf{X}^T\hat{\mathbf{L}}\mathbf{X})$ is equal to $\frac{1}{2}\sum_{i,j=1}^N \mathbf{S}_{ij}(\frac{\mathbf{x}_i}{\sqrt{\mathbf{D}_{ii}}} - \frac{\mathbf{x}_j}{\sqrt{\mathbf{D}_{jj}}})^2$. Since degrees of the perturbed

Table 1: Description of datasets

|  | $N_{LCC}$ | $E_{LCC}$ | Classes | Features |
|---|---|---|---|---|
| Cora | 2485 | 5069 | 7 | 1433 |
| Citeseer | 2110 | 3668 | 6 | 3703 |
| Cora-ML | 2810 | 7981 | 7 | 2879 |
| Polblogs | 1222 | 16714 | 2 | / |

graph are approximately equal to those of the real graph, for the convenience of calculation, we let $\mathbf{D}_{ii} = \sum_{j=1}^{N} \widetilde{\mathbf{A}}_{ij}$. $\|\mathbf{S}\|_* = \sum_{i}^{rank(\mathbf{S})} \sigma_i$, where $\sigma_i$ is the $i$-th singular value of $\mathbf{S}$.

To solve formula(3), we let $\frac{\partial \mathcal{L}1 + \mathcal{L}2}{\partial \mathbf{S}} = 0$ to get the closed form solution.

$$\frac{\partial \mathcal{L}1 + \mathcal{L}2}{\partial \mathbf{S}} = 2(\mathbf{S} - \widetilde{\mathbf{A}}) + c \begin{pmatrix} (\frac{\mathbf{x_1}}{\sqrt{\mathbf{D}_{11}}} - \frac{\mathbf{x_1}}{\sqrt{\mathbf{D}_{11}}})^2 & \cdots & (\frac{\mathbf{x_1}}{\sqrt{\mathbf{D}_{11}}} - \frac{\mathbf{x_N}}{\sqrt{\mathbf{D}_{NN}}})^2 \\ \vdots & \ddots & \vdots \\ (\frac{\mathbf{x_N}}{\sqrt{\mathbf{D}_{NN}}} - \frac{\mathbf{x_1}}{\sqrt{\mathbf{D}_{11}}})^2 & \cdots & (\frac{\mathbf{x_N}}{\sqrt{\mathbf{D}_{NN}}} - \frac{\mathbf{x_N}}{\sqrt{\mathbf{D}_{NN}}})^2 \end{pmatrix} = 0 \qquad (4)$$

$$\mathbf{S} = \widetilde{\mathbf{A}} - \frac{c}{2} \begin{pmatrix} (\frac{\mathbf{x_1}}{\sqrt{\mathbf{D}_{11}}} - \frac{\mathbf{x_1}}{\sqrt{\mathbf{D}_{11}}})^2 & \cdots & (\frac{\mathbf{x_1}}{\sqrt{\mathbf{D}_{11}}} - \frac{\mathbf{x_N}}{\sqrt{\mathbf{D}_{NN}}})^2 \\ \vdots & \ddots & \vdots \\ (\frac{\mathbf{x_N}}{\sqrt{\mathbf{D}_{NN}}} - \frac{\mathbf{x_1}}{\sqrt{\mathbf{D}_{11}}})^2 & \cdots & (\frac{\mathbf{x_N}}{\sqrt{\mathbf{D}_{NN}}} - \frac{\mathbf{x_N}}{\sqrt{\mathbf{D}_{NN}}})^2 \end{pmatrix} \qquad (5)$$

We denote formula(5) as $\mathbf{S} = \widetilde{\mathbf{A}} - \frac{c}{2}\mathbf{Z}$ for convenience. A proximal operator of nuclear norm is adopted to remove noises and reserve main properties(Entezari et al., 2020).

$$\text{prox}_{*\beta}(\mathbf{S}) = \mathbf{U}diag(\max\{\sigma_i - \beta, 0\})_i \mathbf{V}^T \qquad (6)$$

where $\mathbf{S} = \mathbf{U}diag(\sigma_1 \ldots \sigma_N)\mathbf{V}^T$ is the singular value decomposition of $\mathbf{S}$. Let $\mathbf{S} = \text{prox}_{*\beta}(\mathbf{S})$ to represent this step. For the constraint $\mathbf{S}_{ij} \in [0, 1]$, we let $\mathbf{S} = \mathbf{S} + \mathbf{I}$ to enhance self-loop, and set $\mathbf{S}_{ij}<0$ to 0 and $\mathbf{S}_{ij}>1$ to 1. We denote this step as $\mathbf{S} = \text{prox}_s(\mathbf{S})$, which can make the graph sparse at the same time.

### 3.4 THE CONVOLUTION OPERATION FOR FEATURES

After getting the cleaned graph through several graph reconstruction operations above, we fix it and focus on denoising features, formula(1) of GUGNN can be rewritten as follows:

$$\underset{\mathbf{F}}{argmin}\, \mathcal{L} = \|\mathbf{F} - \mathbf{X}\|_F^2 + \lambda \cdot tr(\mathbf{F}^T \mathbf{L} \mathbf{F}) \qquad (7)$$

where $\lambda = \frac{c}{\gamma}$. In this case, formula(7) is equal to that of UGNN(Ma et al., 2020). We use the normalized Laplacian matrix $\hat{\mathbf{L}}$ and let $\frac{\partial \mathcal{L}}{\partial \mathbf{F}} = 0$ to find the optimal solution.

$$\frac{\partial \mathcal{L}}{\partial \mathbf{F}} = 2(\mathbf{F} - \mathbf{X}) + 2\lambda \hat{\mathbf{L}} \mathbf{F} = 0 \qquad (8)$$

$$\mathbf{F} = (\mathbf{I} + \lambda \hat{\mathbf{L}})^{-1} \mathbf{X} \qquad (9)$$

Formula(9) is the process of feature aggregation. Before it, we let $\mathbf{X} = \mathbf{X}\mathbf{W}$ to transform features, where $\mathbf{W}$ is the parameter of a single GNN convolution layer. This is our convolution operation for features, which is proved(Ma et al., 2020) equal to PPNP(Klicpera et al., 2019). So, our convolution operation for features can also adjust nodes' neighborhood to enhance the model's robustness.

### 3.5 THE DESIGN OF R-GUGNN MODEL

Utilizing the two kinds of operations, we design R-GUGNN model and show it in Figure 1, where $m$ is the number of graph reconstruction operations we can set. In step ①, we fix features and clean the graph with $m$ graph reconstruction operations. In step ②, we fix the cleaned graph and denoise features with two graph convolution layers. We train GNN parameters $\theta$ using the two graph convolution layers and classify nodes finally. Concrete steps of R-GUGNN are shown in Algorithm 1.

Table 2: Node classification performance (Accuracy±Std) under *metattack*

| Datasets | Ptb Rate(%) | GCN | GAT | RGCN | GCN-SVD | Pro-GNN | R-GUGNN |
|---|---|---|---|---|---|---|---|
| Cora | 0 | 83.06±0.52 | 84.09±0.66 | 83.83±0.59 | 77.69±0.52 | **85.49±0.38** | 82.97±0.33 |
| | 5 | 77.08±1.05 | 79.96±1.01 | 79.21±0.42 | 77.54±0.91 | 79.03±1.31 | **82.25±0.51** |
| | 10 | 70.46±1.14 | 74.82±1.33 | 73.11±0.76 | 72.73±0.90 | 74.11±0.77 | **80.91±0.30** |
| | 15 | 65.21±1.64 | 70.17±1.34 | 68.68±0.80 | 69.11±0.67 | 70.34±0.50 | **80.60±0.37** |
| | 20 | 54.69±1.37 | 58.33±1.49 | 58.35±0.39 | 57.46±2.04 | 67.78±0.48 | **77.96±1.15** |
| | 25 | 49.53±1.47 | 52.22±2.76 | 53.27±0.56 | 54.46±1.60 | 66.19±0.85 | **76.05±1.37** |
| Citeseer | 0 | 72.22±0.49 | 72.87±1.38 | 72.98±0.29 | 67.89±0.68 | 72.68±0.78 | **73.59±0.33** |
| | 5 | 69.76±1.91 | 71.87±1.60 | 71.66±0.43 | 68.44±0.44 | 72.17±1.67 | **73.15±0.71** |
| | 10 | 67.25±1.30 | 70.02±1.18 | 69.17±0.57 | 69.73±0.88 | 73.06±0.50 | **73.96±0.42** |
| | 15 | 63.87±1.47 | 67.30±2.03 | 65.93±0.37 | 68.06±0.45 | 71.24±0.54 | **73.25±0.80** |
| | 20 | 56.00±1.36 | 60.08±1.09 | 56.83±0.54 | 68.71±0.65 | 69.22±0.65 | **71.64±0.59** |
| | 25 | 57.10±2.45 | 61.00±1.99 | 58.69±0.47 | 65.43±1.01 | 57.23±1.22 | **71.74±0.99** |
| Cora-ML | 0 | 85.77±0.32 | 85.46±0.51 | **85.97±0.42** | 78.78±0.17 | 85.30±0.66 | 85.29±0.24 |
| | 5 | 80.01±0.42 | 81.20±0.81 | 80.68±0.39 | 77.92±0.22 | 83.92±0.45 | **84.70±0.35** |
| | 10 | 74.51±0.56 | 75.97±0.90 | 74.70±0.76 | 77.61±0.39 | 81.69±0.42 | **84.12±0.16** |
| | 15 | 54.36±0.66 | 57.80±1.24 | 55.86±1.06 | 74.92±0.33 | 53.88±0.45 | **82.13±0.46** |
| | 20 | 45.64±0.71 | 42.02±2.36 | 48.08±0.29 | 51.01±0.94 | 46.99±2.82 | **70.74±0.90** |
| | 25 | 48.20±1.45 | 46.68±2.51 | 50.58±0.42 | 66.57±0.51 | 50.82±0.45 | **74.30±0.47** |
| Polblogs | 0 | **95.83±0.40** | 94.99±0.41 | 95.30±0.25 | 91.93±0.37 | 95.25±0.14 | 95.68±0.31 |
| | 5 | 72.81±0.91 | 76.69±0.96 | 72.04±0.54 | 89.10±0.35 | 93.53±0.47 | **95.30±0.57** |
| | 10 | 72.71±0.80 | 72.56±1.33 | 71.89±0.51 | 81.25±0.50 | 87.53±0.83 | **94.37±0.85** |
| | 15 | 68.35±0.42 | 54.73±8.66 | 68.66±0.66 | 70.27±2.42 | 85.88±1.79 | **91.44±5.74** |
| | 20 | 59.34±2.45 | 50.07±4.35 | 62.14±0.80 | 58.73±3.64 | 77.05±3.35 | **84.13±5.08** |
| | 25 | 58.39±1.61 | 50.91±2.45 | 59.89±0.96 | 53.03±2.32 | 70.34±2.05 | **70.72±7.27** |

## 4 EXPERIMENTS

In this section, we evaluate the effectiveness of R-GUGNN model compared with the state-of-the-art GNN models against different attacks. We first introduce the experimental settings and then present results of a series of experiments. At last, we conduct the ablation study and analyze hyper-parameters of R-GUGNN.

### 4.1 EXPERIMENTAL SETTINGS

#### 4.1.1 DATASETS

We compare different models on four benchmark datasets, including three citation graphs, i.e., Cora(McCallum et al., 2000), Citeseer(Giles et al., 1998) and Cora-ML(Bojchevski & Günnemann, 2017), and one blog graph, i.e., Polblogs(Jin et al., 2020b). Cora-ML is the subset of machine learning papers from Cora dataset, which is also a well-known dataset in GNN field. Since Polblogs dataset has no node features, a $N \times N$ identity matrix is used to act as the feature matrix. We only consider the largest connected component(LCC) in each dataset(Jin et al., 2020b; Zügner et al., 2018). Table 1 contains detailed information about the dataset.

#### 4.1.2 BASELINES

R-GUGNN model are compared with the state-of-the-art GNN and defense models in repository DeepRobust(Li et al., 2020), i.e., GCN(Kipf & Welling, 2017), GAT(Velickovic et al., 2018), RGCN(Zhu et al., 2019), GCN-SVD(Entezari et al., 2020) and Pro-GNN(Jin et al., 2020b). We adopt the default parameter settings in GCN and GAT. The number of hidden units of RGCN are tuned from {16, 32, 64, 128}. The reduced rank of the perturbed graph in GCN-SVD is tuned from {5, 10, 15, 50, 100, 200}. For Pro-GNN, we use the tuned hyper-parameters the author gives online.

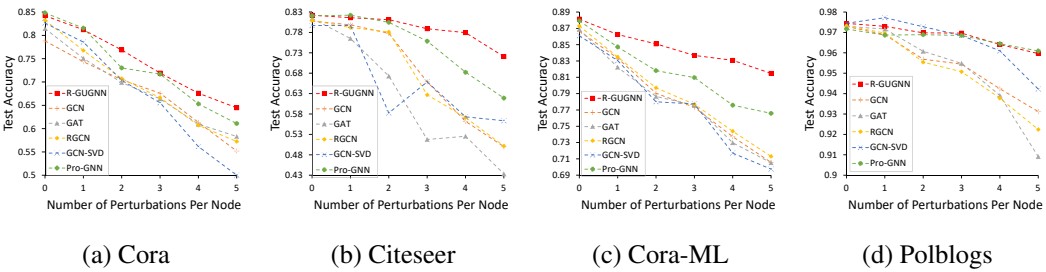

Figure 2: Node classification performance (Accuracy) under $nettack$

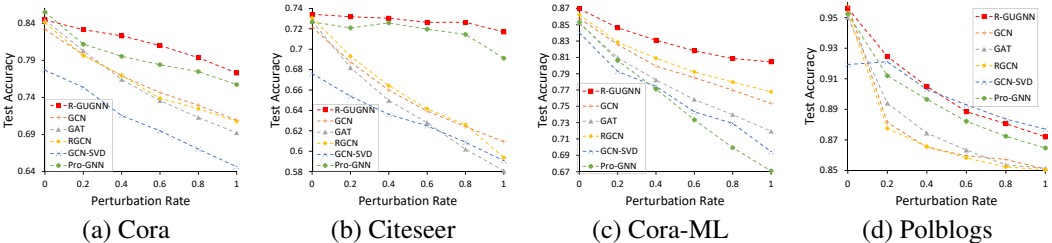

Figure 3: Node classification performance (Accuracy) under random attack

### 4.1.3 PARAMETER SETTINGS

Just as(Jin et al., 2020b), for each dataset, we choose 10% of nodes for training, 10% of nodes for validation and the remaining 80% of nodes for testing. The average performance of 10 runs is reported for all experiments below. The hyper-parameters of all the models are tuned based on the loss and accuracy on the validation set. Note that the same hyper-parameters are used under the same attack for the same dataset no matter what perturbation rate is. If there are no special instructions, all models adopt two graph convolution layers with 16 hidden units. Learning rate of Adam optimizer $\eta$ is fixed as 0.01 and negative log likelihood loss is adopted for a fair comparison(RGCN has its own loss function).

### 4.2 PERFORMANCE AGAINST DIFFERENT ATTACKS

The node classification performance of R-GUGNN is evaluated against three types of poisoning attacks, i.e., global attack, targeted attack and random attack. Since Ploblogs dataset has no real node features, hyper-parameter $c$ of R-GUGNN is set to 0 on Ploblogs dataset.

### 4.2.1 AGAINST GLOBAL ATTACK

The famous $metattack$(Zügner & Günnemann, 2019) is used as the global attack to conduct experiments and all the default parameter settings in the authors' original implementation are adopted. Concretely, the strongest variant Meta-Self is applied for all datasets. The perturbation rate of $metattack$ on the graph is from 0% to 25% with a step of 5%, since too heavy attacks are noticeable and make no sense. We report the average accuracy of node classification with standard deviation on test set and highlight the optimal results in bold. Concrete results are shown in Table 2 and we draw some conclusions:

- R-GUGNN has great improvement compared to others on four datasets. The average improvement of accuracy under different perturbation rates over GCN on four datasets is about 16%, 10%, 19% and 21% respectively. When the graph is heavily perturbed, the improvement is larger. For example, when the perturbation rate is 20%, the improvement over GCN is about 23%, 16%, 25% and 25% on four datasets respectively. When compared with different second best models, improvement can reach 10%, 6%, 19% and 7% on four datasets. These results prove that R-GUGNN can defend against $metattack$ very well.

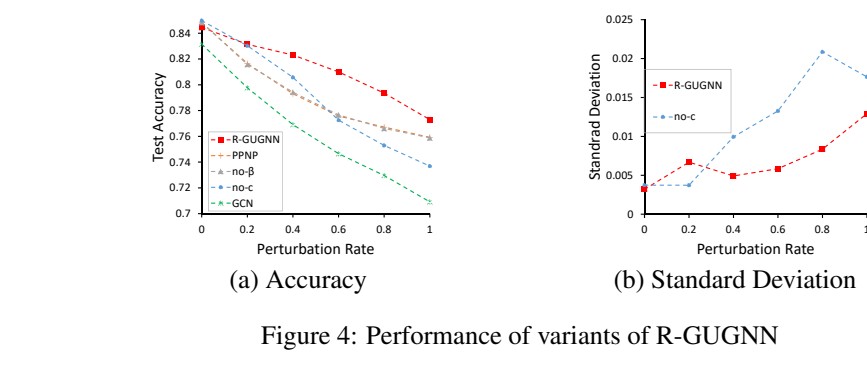

(a) Accuracy            (b) Standard Deviation

Figure 4: Performance of variants of R-GUGNN

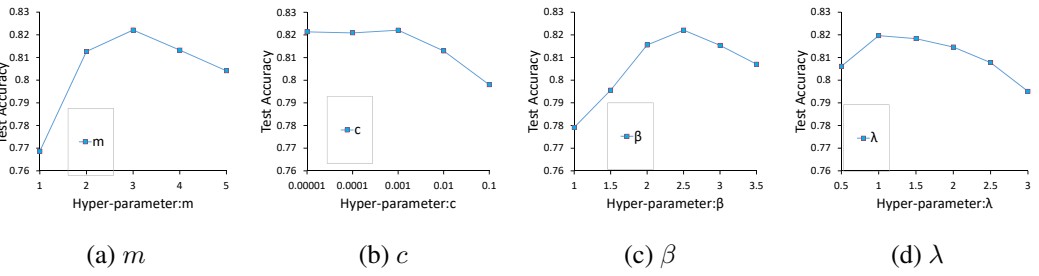

(a) $m$       (b) $c$       (c) $\beta$       (d) $\lambda$

Figure 5: Parameter analysis on Cora-ML dataset under 15% $metattack$

- Accuracy of R-GUGNN is stably high under different perturbation rates on all datasets. The gap between accuracy of R-GUGNN on clean and perturbed graphs is small. For example, gaps of the accuracy of R-GUGNN under 25% and 0% $metattack$ on Cora and Citeseer datasets are only about 7% and 2%. Besides, the overall standard deviations of R-GUGNN are small. However, the lack of real node features on Polblogs dataset causes big standard deviations when the graph is heavily attacked.

#### 4.2.2 AGAINST TARGETED ATTACK

The typical $netttack$(Zügner et al., 2018) is employed as the targeted attack to conduct experiments and all the default parameter settings in the authors' original implementation are adopted. We select nodes with degree $>10$ as targeted nodes from the test set. The number of perturbations of the graph on each targeted node is from 0 to 5 with a step of 1. We report accuracy of these targeted nodes as results, which are shown in Figure 2. R-GUGNN suffers less effects of $netttack$ and also performs greatly and stably. For example, compared to the second best method Pro-GNN, R-GUGNN achieves 10% and 5% improvement on Citeseer and Cora-ML datasets. These results prove that R-GUGNN can defend against $netttack$ very well.

#### 4.2.3 AGAINST RANDOM ATTACK

Performance of R-GUGNN under random attack is evaluated here. We add random perturbations on the graph from 0% to 100% with a step of 20%. Concrete results are shown in Figure 3. R-GUGNN outperforms other models again and the improvement is distinct. For example, compared to different second best models, R-GUGNN achieves a 2.5% and 3.5% improvement on Citeseer and Cora-ML datasets. These results prove that R-GUGNN can defend against random attack very well.

From the overall performance, we observe that the advantage of R-GUGNN is obvious compared with others and its performance is stably great. In conclusion, R-GUGNN is robust enough to defend against different attacks.

### 4.3 ABLATION STUDY

R-GUGNN contains $m$ graph reconstruction operations. If we discard these operations and only use two convolution layers for features, R-GUGNN is equal to PPNP(Klicpera et al., 2019). So, in this

subsection, we compare PPNP with GCN and R-GUGNN on Cora dataset under random attack as an example to illustrate. In addition, we set $\beta$ and $c$ to 0 to understand the impact of each component in the graph reconstruction operation. Furthermore, we observe standard deviations of R-GUGNN when $c$=0.

In Figure(4)(a), we can see that performance of R-GUGNN is better than PPNP and performance of GCN is the worst. It shows graph reconstruction operations are significant(R-GUGNN vs PPNP), and adjusting neighborhood is beneficial to defending against attacks(PPNP vs GCN). PPNP curve and the no-$\beta$ curve overlap very well, which indicates that removing noises with small singular values plays a quite important role in cleaning the graph. What's more, if $c$=0, not only model's performance is poor, but also the standard deviation rises a lot especially when the graph is heavily attacked in Figure(4)(b). It shows the Laplacian regularization term is significant in improving stability of R-GUGNN, which explains why the standard deviation is big on Polblogs dataset under heavy attack.

### 4.4 PARAMETER ANALYSIS

In this subsection, we show performance of R-GUGNN with different values of hyper-parameters i.e., $m$, $c$, $\beta$, and $\lambda$. We use Cora-ML dataset under 15% $metattack$ as an example to illustrate. The value range of $m$ is from 1 to 5 with the step of 1. The value of $c$ is selected in $\{10^{-5}, 10^{-4}, 10^{-3}, 10^{-2}, 10^{-1}\}$. We select $\beta$ from 1 to 3.5 and $\lambda$ from 0.5 to 3 with the step of 0.5. In the process of tuning one hyper-parameter, other hyper-parameters are fixed as the optimal. Figure 5 shows effects of different values of hyper-parameters.

$m$ is the number of graph reconstruction operations. Our novel operations of R-GUGNN are important for defending against attacks, and even one such operation improves model's robustness(76.87% accuracy). However, proper $m$ can boost the accuracy and too many such operations cannot benefit R-GUGNN. $\beta$ is also a key affecting the performance of R-GUGNN, which controls how many noises with small singular values to remove. When $\beta$ is too small, noises cannot be removed entirely. While when $\beta$ is too big, the main properties of the graph can be hurt. $\lambda$ is used to adjust the size of nodes' neighborhood when propagating features and choosing proper $\lambda$ is also important. $c$ is used to control the Laplacian smoothness of the graph. We find the big value of $c$ hurts the performance of R-GUGNN, but when $c$ is small, accuracy doesn't decrease a lot. From a whole performance, all hyper-parameters have an interval of values where the performance of R-GUGNN is stably great.

## 5 CONCLUSION

In this paper, we propose GUGNN, a novel general unified framework to effectively enhance the robustness of GNNs against adversarial attacks by jointly cleaning the perturbed graph and denoising the features of data. Furthermore, we extend this framework by reconstructing the graph and making convolution operations of features with intrinsic properties, and propose a robust GNN model R-GUGNN. Experiment results show that R-GUGNN stably outperforms the state-of-the-art baselines under different adversarial attacks. In the future, we aim to extend this framework to other models on graphs, even more complicated graph structures for mining the rich value underlying graph data of various domains.

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
