# OpenReview forum: "A General Unified Graph Neural Network Framework Against Adversarial Attacks"
_ICLR.cc/2022/Conference — ICLR 2022 Submitted_

### Official Review · Reviewer_4H1G · 2021-11-02

**Correctness:** 4
**Technical Novelty And Significance:** 2
**Empirical Novelty And Significance:** 3
**Recommendation:** 5
**Confidence:** 4

**Main Review:**

Positives:

+ The paper is well-written.
+ The problem is relevant. In particular, the idea of learning how to fix the perturbed structure as well as the node features is interesting.
+ The results section is also well designed with three attack models, ablation studies, and hyperparameter results.

 Negatives:

- The novelty of the paper is not clear. Part of the objective (Eq 1) is taken from UGNN (Ma et al. 2020). Also, the approach taken in the paper is step-wise. After denoising the graph through reconstruction operations, it is solving Eq. 7 (same as in UGNN).  As a backup clarification, why isn't UGNN being used as a baseline (or in the ablation study)?

 - It would be good to have a comparison with other defense architectures. There are already methods that defend against poisoning attacks (e.g., GNNGuard, NeurIPS, 2020). Pro-GNN does not seem to be the state of the art.

- About the reconstruction: It is not clear what should be an optimal number of rounds for reconstruction. It feels ad hoc as of now. The results on m (number of rounds) are also not convincing. Why should the results be worse for a higher value of m? It will be good to have some discussion around all of these.

Minor:

- It will be good to have a discussion about the efficiency of the proposed method.

- The model seems to produce a high standard deviation for polblogs data. I am not sure if one can project R-GUGNN as the best model for that dataset. Is there any intuition/explanation for why it is happening?

**Summary Of The Paper:**

The paper proposes a robust model for graph neural networks (GNNs) to defend against adversarial attacks. While most of the defense mechanisms in previous work focus on identifying and fixing the perturbed structure of the underlying graph, this paper proposes to jointly fix the perturbed structure as well as the node features. The proposed objective function captures these components along with weighting by some hyper-parameters. The experiments include several baselines and show results on different attack models: targeted attack, untargeted or global attack, and random attack. The experiments also have results on ablation study as well as hyper-parameters.

**Summary Of The Review:**

Please refer to the negative points in the detailed review.

---

> ### Author Response · Authors · 2021-11-22
> **Thank you for your comments.**
>
> Thank you for your comments. We refer to some contents of UGNN and Pro-GNN. As the performance of UGNN is reportedly similar to that of PPNP, so we only compare R-GUGNN with PPNP in the ablation experiment and do not compare it with UGNN
> Thanks for your recommendation on GNNGuard. In the future, we will further take GNNGuard as one of the baselines. The number of epochs of graph purification m is really unclear in theory. We will rethink the framework and try to further improve the model. In addition, we will minimize the variance of the results in the future. Thank you for your comments again.

---

> > ### Comment · Reviewer_4H1G · 2021-11-26
> > **View on Rebuttal**
> >
> > Thank you for your reply. I appreciate the positivity in your rebuttal and will look forward to an updated model and its corresponding results.

---

### Official Review · Reviewer_Hj7t · 2021-11-02

**Correctness:** 3
**Technical Novelty And Significance:** 2
**Empirical Novelty And Significance:** 2
**Recommendation:** 3
**Confidence:** 5

**Main Review:**

Strengths:
- The paper tackles an important problem
- The paper is easy to follow
- Ablation studies and various experiments show the potential benefit of the technique (though, the datasets are rather small and further competitors could be used)

Weaknesses:
- Technical novelty and model definition: I am not fully convinced by the approach. In particular, the authors first perform "graph cleaning" and only subsequently denoise the features. Not only, leads this to a low technical novelty (since graph cleaning has been proposed already; likely the feature denoising -- as mentioned by the authors -- is actually PPNP) but also to some questionable choices.
In particular, in Section 3.3. the authors assume F=X (i.e. the features are NOT perturbed). This is a strong contraction to the core idea that the features are actually perturbed. Accordingly, the graph cleaning step clearly is based on some "wrong" features.
Moreover, the claim in the abstract of "simultaneously consider denoising features" is also not really solved by the proposed technique, since it is only sequentially considered.
- Model definition (II): In general, the model definition is unclear and/or not well presented. For example, Eq. (1) does not contain any learnable parameters and only considers the "cleaning". Only later it is added that X is actually replaced by some learnable transformation. This also leads to the situation, that learning actually only happens for the "feature part" (second part in Algo 1) but not the graph part. I find this very unsatisfying. Or put differently: The graph cleaning could then be used in combination with ANY standard GNN. It would be interesting to include such a comparison.
- Model definition (III): The assumption that the degrees of the perturbed and cleaned graphs are the same seems very rough. Also, it is unclear to me why one considers the derivative of L1+L2, even though the full objective is L1+L2+L3.
- Experiments: A comparison on more complex datasets and with more competitors would be helpful. In particular, as mentioned above, the graph cleaning could be used with any GNN. Is the actual benefit of the proposed work the graph cleaning? Since the feature part corresponds to PPNP, it seems to be the case. It would be helpful is the authors discuss this point more carefully.

Further (minor) points:
- The abstract/introduction reads a bit like "no work has considered feature perturbation". This is, however, incorrect. E.g., as mentioned by the authors, Netattack actually consideres features. Also works like "Certifiable Robustness and Robust Training for Graph Convolutional Networks, KDD'19" do consider robustness regarding feature perturbations. I would recommend to rephrase the corresponding paragraphs.



**Summary Of The Paper:**

Considering the problem of adversarial manipulations of graphs, the authors propose a framework for "cleaning" the graph structure and its features to obtain more robust predictions. The core contribution, compared to existing works, is the additional considering of the feature cleaning. Technically, the authors first perform the graph cleaning (phrased as some kind of sparse, trace minimization problem) followed by some feature cleaning (phrased as some kind of feature diffusion/propagation). Experiments on standard (small) benchmark datasets shows some improvements.

**Summary Of The Review:**

The model definition of the paper is not convincing and the experimental results needs a more careful investigation. In particular, the novelty seems to be limited to the graph cleaning since the remaining parts are an existing model (PPNP).

---

> ### Author Response · Authors · 2021-11-22
> **Thank you for your comments.**
>
> Thank you for your comments. As what you said, we deal with graphs and features sequentially. We will continue to improve in the future. We regard the information aggregation of the convolution part of the graph as a denoising process, but in fact we don't perturb the features. Thus, we first utilize features to play a role in the purification of the graph.
> We will later apply this framework to more GNN models to prove its universality. Thank you again for your comments. We will improve our method completely later.

---

> > ### Comment · Reviewer_Hj7t · 2021-11-26
> > **Thank you**
> >
> > Thank you for your reply. I am looking forward to an updated version!

---

### Official Review · Reviewer_z4si · 2021-11-02

**Correctness:** 2
**Technical Novelty And Significance:** 2
**Empirical Novelty And Significance:** 3
**Recommendation:** 5
**Confidence:** 4

**Main Review:**

**Problem and motivation**

The robustness of graph-based machine learning models is an important problem worthy of study. The authors do a good job explaining the problem and motivating its importance.

They also highlight the lack of research jointly considering feature and structural robustness, rather than just structural robustness. Although it should be noted that the Pro-GNN framework does consider both the graph and the features [2].

On adversarial attacks, the authors state "They always try to add, delete, or rewire edges to change the graph structure". This is not true, some do modify features too (such as nettack) and some also change the graph structure in other ways (i.e. through node injection [1])

**Adversarial attack taxonomy**

Throughout the work, there seems to be some confusion about adversarial attack taxonomy. For example, the authors write “Poisoning attacks have various types, including global attack, targeted attack and random attacks”. Whilst true, having a global or targeted setting is not unique to poisoning attacks, the other common setting of evasion attacks can also be global or targeted. Meanwhile, a random attack is a strategy of attack and can be applied in any adversarial setting. In 4.2 the authors write “.. R-GUGNN is evaluated against three types of poisoning attacks, i.e., global attack, targeted attack and random attack.”. Again, global and targeted attacks are an attack goal and random attack is a strategy. The authors may find Section 3 of [3] helpful. Related to this, the sentence “Random attack adds random noises to the clean graph whose concrete practice is adding, removing or flipping edges randomly”, this sentence is not easy to parse and it’s unclear what the authors are trying to say.

**Framework**

The GUGCN framework provides a cleaned version of the adjacency matrix and the feature matrix. The terms promote the following properties:
+ Provide a cleaned adjacency that is not too far from the original
+ Provide a feature matrix that is not too far from the original
+ Provide a feature matrix that is smooth with respect to the learned cleaned matrix
+ Provide a cleaned adjacency according to some prior f(S)

The authors claim the framework is general and unified, however, I have some concerns about this claim:
+ The term which is concerned with the feature matrix promotes a specific property (smoothness) whereas the term concerned with the adjacency is general. It’s unclear why one is specific and the other is general.
+ Despite having a general term for the adjacency matrix, the authors only consider a single realisation (low-rank regularisation), without suggestions for other possibilities.
+ I believe that if the authors want to propose a general and unified framework they should show that existing works are a special case of the framework or provide more than one instantiation of the framework. The authors do show that PPNP is a are special case, but this is given late in the paper in the results section (Section 4.3).
+ The authors promote the smoothness of features, which is obviously preferable for homophilic datasets. I doubt this method would work well on non-homophilous datasets. Since most of the current benchmark semi-supervised node classification datasets are homophilous this isn’t a major issue,  but the authors could maybe acknowledge this point in their manuscript.

I believe this work could be greatly improved if they demonstrated exactly how previous works fit into this framework. Currently, it is unclear what their contribution is over other works, as it seems similar to some existing works. Section 3 which derives steps 1-5 of Algorithm 1 look like it’s the same as equation (17) of the Pro-GNN paper [2]. However, I don’t think this connection is made in the manuscript. The final objective of Pro-GNN (Eq. 9) is also similar to the GUGCN objective.

The naming is quite confusing, as the GUGCN by itself is not a graph convolutional network.

I’m unsure of the meaning of the following sentence “Although X and A have some noises, they can represent the real features and the graph to a large extent”.

**Derivation**
Moving from Eq (2) to Eq (3) the authors switch the framework to operating on an adjacency matrix and the combinatorial Laplacian to an adjacency matrix with self-loops and normalised Laplacian matrix. This change is not discussed or justified, and Eq (3) no longer fits into the GUGNN framework as defined in Eq (1)

I don’t think the equation at the bottom of page 4 should have the (1/2) term at the front.

"Since degrees of the perturbed graph are approximately equal to those of the real graph, for the convenience of calculation, we let". The authors here seem to have made an assumption on the perturbation they are considering and should be more clear about this. Furthermore, the convenience they make is stronger than what they claim, the degrees are not approximately equal they are exactly equal.

Section 3.3 in general, particularly the last paragraph, is quite confusing and lacks details and references. It is not clear to me where the final term in Eq. (4) comes from.

The authors should comment on the scalability of Eq. (9) which is equivalent to PPNP. Due to the need for inverting a matrix, this method is unlikely to scale to networks with 100,000+ nodes. The authors could discuss using APPNP instead.

**Experiments**

The experiments section is the strongest part of this paper. It is very thorough and includes key analyses such as ablation studies and sensitivity to hyper-parameters. The results are convincingly improved compared to baseline methods.

The standard deviations in Figure 4 (a) are small, could they be added to Figure 4 (b) as error bars?

In Section 4.4 the ranges for each hyper-parameter are a bit inconsistent and the authors do not discuss why this is. Why is c chosen on a log-scale but the other parameters aren't? Why are the other parameters (beta and lambda) not chosen on the same linear scale (e.g. 0.5, ..., 3.5 in steps of 0.5)?

The layout of results is similar to Pro-GNN (table 2, figure 2 and figure 3 in this paper vs table 2, figure 3 and figure 4 in Pro-GNN), yet the results in some cases for Pro-GNN are quite different. For example at 25% Ptb rate on Cora Jin et al. report 69.72 ± 1.69 whereas this paper reports 66.19 ± 0.85. For Citeseer Jin et al. report 68.95 ± 2.78 whereas this paper reports 57.23 ± 1.22. Do the authors know why these are so different?

From the code, it looks like although the model training is done over many random seeds, the same poisoned graph is used each time. The variance estimates would be more reliable if a different perturbed graph was used for each seed. I believe authors should change this, or at least mention it in the manuscript.

**Minor points and formatting suggestions**
The formatting could be improved in places. For example:
+ No space between acronyms “Graph Neural Networks(GNNS)”
+ Multiple spaces after full stops “among nodes.   Graphs”
+ No space between citations “systems(Ying et al. 2018a)”
+ R -> \mathbf{R} for set of reals
+ Netttack -> nettack

**Code (not factored into my scores)**
The code in the supplementary does not run. Trying to run api.py will give an import error. It looks like “from attacked_data import PrePtbDataset” should be changed to “from deeprobust.graph.attacked_data import PrePtbDataset”. The api.py script could be improved by using argparse.

[1]: Tao, Shuchang, et al. "Single Node Injection Attack against Graph Neural Networks." arXiv preprint arXiv:2108.13049 (2021).

[2]: Jin, Wei, et al. "Graph structure learning for robust graph neural networks." Proceedings of the 26th ACM SIGKDD International Conference on Knowledge Discovery & Data Mining. 2020.

[3]: Sun, Lichao, et al. "Adversarial attack and defense on graph data: A survey." arXiv preprint arXiv:1812.10528 (2018).






**Summary Of The Paper:**

The authors propose a General Unified Graph Neural Network (GUGNN) framework to address graph and feature denoising. They use this framework to design a graph neural network (R-GUGNN) which is Robust against poison adversarial attacks in a semi-supervised node classification setting. The authors demonstrate on four common datasets (Cora, Citeseer, Cora-ML & Polblogs) that their method is generally more robust against perturbation compared to commonly used graph neural networks (GCN, GAT) and models which are designed to be robust in the same setting (RGCN, GCN-SVD, Pro-GNN).

**Summary Of The Review:**

The authors propose an interesting idea, to jointly sanitise the graph structure and the features as a defence against adversarial attack. The experiments are extensive and thorough and the method appears to perform well. However, the paper is let down by two major factors. The first is that it is confusing how their framework is general and how it relates to prior works. It’s also not obvious specifically how they are innovating compared to prior works as parts of their method are similar to some previous studies. The second is that the derivation is lacking details and references and, as it is written, is not at all easy to follow. I believe this paper has potential but in its current state, is not yet ready to be published.

---

> ### Author Response · Authors · 2021-11-22
> **Thank you for your comments.**
>
> Thank you for your comments. We use PPNP to deal with features without showing the effects of other GNN models. We will supplement experiments in the future to prove the generality of this framework. We really didn't do experiments on non-homophilous datasets, and we will continue to supplement them in the future.
> What we call generality is for UGNN, which is a framework that can cover a variety of graph convolution operations for features, but it does not take the perturbations of graphs into account. As you suggest, we will continue to improve the framework and add more proof in the future.
> For all baselines, we set the learning rate to 0.01 in all experiments, so the results of Pro-GNN change greatly. We don’t change the other parameters. Finally, thank the reviewers again for their suggestions on the paper, framework, codes, experiments and so on. Thank you very much.

---

### Official Review · Reviewer_V2ZH · 2021-11-02

**Correctness:** 2
**Technical Novelty And Significance:** 2
**Empirical Novelty And Significance:** 2
**Recommendation:** 5
**Confidence:** 5

**Main Review:**

Pros:
1.	This paper focuses on an important problem of learning robust GNN against attacks on both structure and node features.
2.	The proposed method shows good results on four small graphs and perform better than most of the robust GNN methods.
3.	Ablation study and hyperparameter analysis are conducted, which makes it easier to reproduce the results of the proposed method

Cons:
1.	The proposed method lacks novelty. This method proposes to discard the perturbations on graph structure by learning a adjacency matrix S that link nodes with similar node features. And a constraint in Eq(2) is added to S to make sure it will not differ a lot from the poisoned adjacency matrix A. However, these constraints have been proposed by ProGNN[1]. The proposed method simply extends it to graph whose node features are also attacked by adding one more regularizer to ProGNN, which lacks novelty. In addition, this straightforward extension seems problematic. Because the node features are attacked, nodes with similar attacked features may not indicate clean edges anymore. In other words, the graph denoising relies on the node features. Attacked node features may lead to poor graph learning and further feature denoising. It is suggested to have more discussions and justification about the proposed method.

2.	Since the proposed method focuses on adversarial attacks on both features and structures, more details about the experimental settings on attacks such as the levels of perturbations on node features are required.

3.	Experiments are only conducted on four small datasets. It is suggested to conduct experiments on large datasets to show the scalability of the proposed method. It is also suggested to compare with some methods that also cares the attacks on node features such as [2] to make the comparisons more convincing.

[1] Jin, Wei, et al. "Graph structure learning for robust graph neural networks." Proceedings of the 26th ACM SIGKDD International Conference on Knowledge Discovery & Data Mining. 2020.

[2] Feng, Fuli, et al. "Graph adversarial training: Dynamically regularizing based on graph structure." IEEE Transactions on Knowledge and Data Engineering (2019).


**Summary Of The Paper:**

This paper focuses on developing a robust GNN model to defend against adversarial attacks on both graph structures and node features. More specifically, a general unified graph neural network is proposed to learn a graph structure and node features to correct the adversarial attacks. This a two-step approach. First, one operation is applied to reconstruct the graph with the node features. Then, the node features are updated by another operation with the learned graph structure. Experiments are conducted on four small datasets for evaluation.

**Summary Of The Review:**

In summary, this paper studies a robust GNN against adversarial attacks on both graph structure and node features. However, the methodology lacks novelty and requires further justification. Some important technical details are also not covered. Though the experimental results on the small datasets are good, some more experiments on large datasets and the details of the experimental settings are required.

---

> ### Author Response · Authors · 2021-11-22
> **Thank you for your comments.**
>
> Thank you for your comments. We took some ideas from Pro-GNN and SVD-GCN on the low-rank estimate and feature smoothing in graph purity. Besides, considering original features having some noise, we aimed at establishing a framework to optimize features and graphs at the same time, which could be also extended by adding different constraints further. We first use the original features to purify the graph because original features are not perturbed in all experiments. We the aggregation process of features as denosing features as UGNN. Concrete perturbations on the graph are described in Section 4.2. As you suggest, we will apply framework on bigger data sets in further studies. Thank you again for your comments.

---

### Decision · Program_Chairs · 2022-01-20

**Decision:**

Reject

**Comment:**

The paper studies a robust GNN against adversarial attacks on both graph structure and node features.
The reviewers agree that the paper need to improve in terms of novelty and more technical details to meet ICLR standard.